# Whole Genome Characterization and Evolutionary Analysis of G1P[8] Rotavirus A Strains during the Pre- and Post-Vaccine Periods in Mozambique (2012–2017)

**DOI:** 10.3390/pathogens9121026

**Published:** 2020-12-06

**Authors:** Benilde Munlela, Eva D. João, Celeste M. Donato, Amy Strydom, Simone S. Boene, Assucênio Chissaque, Adilson F. L. Bauhofer, Jerónimo Langa, Marta Cassocera, Idalécia Cossa-Moiane, Jorfélia J. Chilaúle, Hester G. O’Neill, Nilsa de Deus

**Affiliations:** 1Instituto Nacional de Saúde (INS), Distrito de Marracuene, Maputo 3943, Mozambique; simone.boene@ins.gov.mz (S.S.B.); assucenio.chissaque@ins.gov.mz (A.C.); adilson.bauhofer@ins.gov.mz (A.F.L.B.); Jeronimo.Langa@ins.gov.mz (J.L.); marta.Cassocera@ins.gov.mz (M.C.); idalecia.moiane@ins.gov.mz (I.C.-M.); jorfelia.chilaule@ins.gov.mz (J.J.C.); nilsa.dedeus@ins.gov.mz (N.d.D.); 2Centro de Biotecnologia, Universidade Eduardo Mondlane, Maputo 3453, Mozambique; 3Instituto de Higiene e Medicina Tropical (IHMT), Universidade Nova de Lisboa, UNL, Rua da Junqueira 100, 1349-008 Lisbon, Portugal; 4Enteric Diseases Group, Murdoch Children’s Research Institute, 50 Flemington Road, Parkville, Melbourne 3052, Australia; celeste.donato@mcri.edu.au; 5Department of Paediatrics, the University of Melbourne, Parkville 3010, Australia; 6Biomedicine Discovery Institute and Department of Microbiology, Monash University, Clayton 3800, Australia; 7Department of Microbial, Biochemical and Food Biotechnology, University of the Free State, 205 Nelson Mandela Avenue, Bloemfontein 9301, South Africa; strydoma@ufs.ac.za (A.S.); oneillhg@ufs.ac.za (H.G.O.); 8Institute of Tropical Medicine (ITM), Kronenburgstraat 43, 2000 Antwerp, Belgium; 9Departamento de Ciências Biológicas, Universidade Eduardo Mondlane, Maputo 3453, Mozambique

**Keywords:** rotavirus group A, G1P[8], whole genome sequencing, Rotarix^®^, Bayesian analysis, Mozambique

## Abstract

Mozambique introduced the Rotarix^®^ vaccine (GSK Biologicals, Rixensart, Belgium) into the National Immunization Program in September 2015. Although G1P[8] was one of the most prevalent genotypes between 2012 and 2017 in Mozambique, no complete genomes had been sequenced to date. Here we report whole genome sequence analysis for 36 G1P[8] strains using an Illumina MiSeq platform. All strains exhibited a Wa-like genetic backbone (G1-P[8]-I1-R1-C1-M1-A1-N1-T1-E1-H1). Phylogenetic analysis showed that most of the Mozambican strains clustered closely together in a conserved clade for the entire genome. No distinct clustering for pre- and post-vaccine strains were observed. These findings may suggest no selective pressure by the introduction of the Rotarix^®^ vaccine in 2015. Two strains (HJM1646 and HGM0544) showed varied clustering for the entire genome, suggesting reassortment, whereas a further strain obtained from a rural area (MAN0033) clustered separately for all gene segments. Bayesian analysis for the VP7 and VP4 encoding gene segments supported the phylogenetic analysis and indicated a possible introduction from India around 2011.7 and 2013.0 for the main Mozambican clade. Continued monitoring of rotavirus strains in the post-vaccine period is required to fully understand the impact of vaccine introduction on the diversity and evolution of rotavirus strains.

## 1. Introduction

Rotavirus is one of the leading causes of diarrheal disease in children under five years of age [1,2]. Worldwide, the number of deaths due to rotavirus infection in children under five years of age in 2016 was estimated to be 128,500, of which 104,733 occurred in sub-Saharan Africa [2]. Rotavirus is a member of the *Reoviridae* family. The genome is comprised of 11 double-stranded ribonucleic acid (dsRNA) segments. The mature virus has an icosahedral capsid formed by three concentric protein layers. The 11 segments of the rotavirus genome encode 12 viral proteins: 6 structural proteins VP1-VP4, VP6 and VP7and 6 non-structural proteins (NSP1-NSP6) [3,4,5,6]. 

The gene segments encoding the external capsid proteins, VP7 and VP4, are used in a binary classification system defining G and P genotypes, respectively [5,7]. Currently, 36 G and 51 P genotypes have been described in humans and various animal species [7,8,9,10]. At least 73 combinations of human rotavirus group A (RVA) G/P genotypes have been described, of which the most common combinations are G1P[8], G2P[4], G3P[8], G4P[8], G9P[8] and G12P[8] [10,11]. However, the implementation of whole genome sequencing has led to comprehensive sequence-based classification of all RVA genes into genotypes, which are identified and differentiated according to particular cut-off values of nucleotide sequence identities [9,11]. Currently, 26 I (VP6), 22 R (VP1), 20 C (VP2), 20 M (VP3), 31 A (NSP1), 22 N (NSP2), 22 T (NSP3), 27 E (NSP4) and 22 H (NSP5) genotypes have been described [8]. The whole genome constellation of a strain can be described following the nomenclature Gx-P[x]-Ix-Rx-Cx-Mx-Ax-Nx-Tx-Ex-Hx. Two major genotype constellations have been designated for strains that commonly infect humans: Wa-like (I1-R1-C1-M1-A1-N1-T1-E1-H1) and DS-1-like (I2-R2-C2-M2-A2-N2-T2-E2-H2). A third constellation also observed in human strains, called AU-1-like (I3-R3-C3-M3-A3-N3-T3-E3-H3), has been shown to have a feline/canine origin [9,11].

Four live oral vaccines, namely Rotarix^®^ (GlaxoSmithKline Biologics, Rixensart, Belgium), RotaTeq^®^ (Merck & Co., Kenilworth, NJ, USA), Rotavac^®^ (Bharat Biotech, Hyderabad, India) and Rotasiil^®^ (Serum Institute of India Pvt. Ltd., Pune, India) have been prequalified by the World Health Organization [12,13]. Rotarix^®^ and RotaTeq^®^ have been introduced into the immunization programs of more than 100 countries [13]. Rotarix^®^ is a monovalent vaccine containing a single human G1P[8] strain and is administered from the age of six weeks [13]. Prior to vaccine introduction in Mozambique, a high burden of rotavirus disease was reported in children under five years old. The rate of rotavirus infection in urban (Maputo City) and rural (Manhiça District) areas between 2012 and 2013 was 42.4% [6]. In 2011, a 24.0% infection rate was reported in the Gaza province, another rural area in southern Mozambique [14]. In both studies G1P[8] was detected at a low frequency [14,15]. Data from the National Surveillance of Diarrhea (ViNaDia) revealed a high rotavirus infection rate of 40.2% and 38.3% in 2014 and 2015, respectively [16]. The Rotarix^®^ vaccine was introduced in Mozambique in 2015 with increasing vaccine coverage of 70% and 80% in 2016 and 2017, respectively [16,17]. Post vaccine introduction, the rotavirus infection rate was reduced to 12.2% and 13.5% in 2016 and 2017, respectively [16]. During ViNaDia surveillance, G1P[8] strains were consistently observed in the pre- (2012–2015) and post-vaccination period (2016–2019). However, in the post-vaccine period a decrease in G1P[8] strains was observed which coincided with the emergence of other non-G1P[8] genotypes such as G3P[4] and G3P[8] [18]. 

The whole genomes of Mozambican G2P[4], G8P[4], G12P[6] and G12P[8] RVA strains from the pre-vaccination period have been described [19,20]. However, there are no reports of the whole genome analyses of G1P[8] strains from Mozambique. To address this, the consensus sequences of 36 G1P[8] strains collected between 2012–2017 from vaccinated and non-vaccinated children were analyzed to investigate the diversity and evolution of G1P[8] strains.

## 2. Results

### 2.1. Genome Constellation

A total of 36 G1P[8] (12 from the pre-vaccine period and 24 from the post-vaccine period) strains were successfully sequenced with an average coverage ranging from 450.0 to 46060.5 per sequence (Appendix A). Complete open reading frames (ORFs) were obtained for 393 of the 396 genome segments analyzed. A partial ORF (99.0%) for segment four of RVA/Human-wt/MOZ/HCN0690/2015/G1P[8] was obtained, while two genome segments (encoding VP2 and VP3, respectively) of RVA/Human-wt/MOZ/HGM0059/2014/G1P[8] could not be determined as insufficient data were generated for these two segments (Appendix A). The genotype constellations were determined and all strains exhibited a Wa-like genetic backbone (G1-P[8]-I1-R1-C1-M1-A1-N1-T1-E1-H1). The nucleotide (nt) identities among Mozambican strains varied from 92.5–100.0% and the comparison between Rotarix^®^ and the 11 genes of the Mozambican strains revealed 84.0–97.9% nt identity (Appendix A).

### 2.2. Phylogenetic Analyses

#### 2.2.1. Sequence Analyses of VP7 and VP4 

The VP7 encoding sequences of the 36 Mozambican G1P[8] strains, collected between 2012 to 2017 from non-vaccinated and vaccinated children (Appendix A), were compared with human rotavirus sequences representing VP7 G1 lineages (I-VII) [21,22,23,24,25,26,27,28]. The Mozambican strains clustered into two distinct lineages, I and II (Figure 1a). The majority of Mozambican strains formed a highly conserved clade, and were closely related to various Indian strains circulating between 2012 and 2013. HGM0544 was moderately divergent to the rest of the strains in the clade sharing 99.2–99.7% nucleotide (nt) identity and 98.8–99.4% amino acid (aa) identity. Strains from the pre- and post-vaccine era were intermingled in lineage II. Only two Mozambican strains from this study clustered in the VP7 lineage I and were more diverse than the 34 strains clustering in lineage II. MAN0033, collected in a rural area in southern Mozambique before vaccine introduction, was closely related to Malawian strains from 2012 and to previously characterized Mozambican strains detected in 2011 [14]. HJM1646, collected in southern Mozambique after vaccine introduction, clustered distinctly and only shared 92.5–92.9% nt and 92.7–93.3% aa identity to the other Mozambican strains. HJM1646 clustered with contemporary Indian strains in a sub-lineage of African and global strains (Figure 1a).

The P[8] encoding sequences of the 36 Mozambican strains were compared with human rotavirus sequences representing the four lineages (I–IV) [21,22,23,24,25,26,27] (Figure 1b). The Mozambican strains clustered in the major P[8] lineage III. Similar to the VP7 tree, the majority of Mozambican P[8] sequences formed a highly conserved clade, and were closely related to various Indian strains circulating between 2012 and 2013. The P[8] encoding sequence of HJM1646 clustered with HGM0544, despite clustering in different VP7 lineages. These two strains were moderately divergent to the rest of the study strains in the Mozambican clade and clustered close to another Mozambican strain, RVA/Human-wt/MOZ/0060a/2012/G12P[8]P[14], which was previously detected in the Manhiça district in southern Mozambique [19]. MAN0033 clustered distinctly to the rest of the Mozambican strains sharing 95.7–96.2% nt and 98.3–98.7% aa identity and was closely related to contemporary Malawian strains isolated in 2012 (Figure 1b, Appendix A).

#### 2.2.2. Sequence Analyses of VP1-VP3 and VP6 

Thirty-three Mozambican strains formed conserved, monophyletic clades that were observed in the VP1, VP2 and VP3 trees, closely related to RVA/Human-wt/IND/CMC00034/2013/G1P[8], and within lineages comprising of contemporary African and global strains (Figure 2a–c). In the VP6 tree, the 35 Mozambican strains clustered together but did not form a discrete monophyletic clade, and were closely related to contemporary Indian G1P[8] strains (Figure 2d). HJM1646 and HGM0544 clustered together in the VP1 tree were moderately divergent from the main Mozambican clade (Figure 2a). In the VP2 tree, HJM1646 clustered close to the monophyletic clade while HGM0544 fell within the Mozambican clade (Figure 2b). In the VP3 tree, these strains clustered together, distinct from the Mozambican clade, adjacent to previously characterized G12P[6] Mozambican strains (Figure 2c) [19]. MAN0033 fell within the same lineage as the main Mozambican clade, showing minor divergence in the VP1 and VP2 tree and more pronounced divergence in the VP3 tree, closely related to contemporary Malawian G1P[8] strains (Figure 2a–c). This strain clustered within a different lineage in the VP6 tree, closely related to the same group of Malawian G1P[8] strains and adjacent to the Mozambican G12P[6] strains (Figure 2d). 

#### 2.2.3. Sequence Analyses of NSP1-NSP5/6

The conserved monophyletic clade, comprised of 33 Mozambican strains, was observed in the NSP1–NSP4 trees, with RVA/Human-wt/IND/CMC00034/2013/G1P[8] interspersed within the clade in the NSP2 tree (Figure 2e–h). HJM1646 and HGM0544 continued to show varied clustering patterns across the trees. In the NSP3 tree these strains clustered together and were divergent from the main Mozambican clade, clustering with Indian strains including RVA/Human-wt/IND/CMC00034/2013/G1P[8], and close to Mozambican G12P[6] strains (Figure 2g). HJM1646 was divergent to the Mozambican clade in the NSP1 and NSP4 trees, but clustered close to the monophyletic clade in the NSP2 tree. HGM0544 clustered with the main Mozambican clade in the NSP1, NSP2 and NSP4 trees. In the NSP5 tree, HJM1646 and HGM0544, along with 33 other Mozambican strains, formed a monophyletic clade that was interspersed with global strains (Figure 2i). MAN0033 clustered distinctly to the rest of the Mozambican strains and was closely related to contemporary Malawian strains isolated in 2012 across these trees (Figure 2e–i). The five G12P[6] [19] Mozambican strains fell within neighboring clusters to MAN0033 in the VP6 and NSP2 trees (Figure 2d,f).

### 2.3. Evolutionary Analysis of VP7 and VP4 Genes 

A randomly subsampled dataset of 378 G1 genes that were representative of global strains temporally and genetically were analyzed (Appendix A). The Mozambican strains detected between 2012 and 2017 shared a common ancestral strain circulating in 2009.9 (95% HPD 2008.1–2010.9). Of the Mozambican strains characterized in this study, 34 clustered within the same lineage and shared a most recent common ancestor in 2011.7 (95% HPD 2011.1–2012.0) and diverged from the closest related Indian strains around the same time. MAN0033 and HJM1646 clustered in the other major lineage present in the tree. MAN0033 and closely related Malawian strains shared a common ancestor in 2010.1 (95% HPD 2008.5–2010.9). These variants, circulating in Malawi, Zambia and Mozambique, diverged from a group of Indian G1P[8] strains in 2001.3 (95% HPD 1998.4–2003.5). HJM1646 was divergent to the other G1 strains from Mozambique in this lineage and shared its most recent common ancestor with Indian strains in 2012.9 (95% HPD 2012.4–2013.0) (Appendix A).

A subsampled dataset of 235 P[8] genes, representative of global strains temporally and genetically, was also analyzed. Thirty-three Mozambican strains characterized in this study clustered within the same lineage and shared a common ancestor in 2013.0 (95% HPD 2012.1–2013.6) and diverged from the closest related Indian strains around 2011.6 (95% HPD 2011.2–2011.9) (Appendix A). The most recent common ancestor of HGM0544 and HJM1646 (that was moderately divergent to the rest of the Mozambican strains in the major clade), was estimated to be 2014.1 (95% HPD 2013.0–2014.9). Clustering in a separate lineage to the other Mozambican G1P[8] strains, MAN0033 diverged from the closest Malawian strain in 2010.9 (95% HPD 2009.5–2011.7) (Appendix A).

### 2.4. Comparative Analysis of Neutralizing Antigenic Epitopes of the VP7 and VP4 Genes of Mozambican Strains and the Rotarix^®^ Vaccine Strain

The rotavirus VP7 protein consists of two antigenic epitopes, 7-1 and 7-2, with 7-1 subdivided into 7-1a and 7-1b [31]. The comparative analysis of the VP7 antigenic epitopes between Mozambican strains and the Rotarix^®^ vaccine strain revealed amino acid substitutions in all three antigenic sites. However, most of the amino acid substitutions were observed in antigenic region 7-2. A total of 30 strains shared conserved amino acid differences at positions N147D and 25 strains at M217I. Sporadic mutations were observed in MAN0033 (unvaccinated) and HJM1646 (fully vaccinated) (S123N, K291R and M217T). The HJM1646 strain contained an additional amino acid substitution at N96S (Figure 3).

Activation of the protein VP4 requires proteolytic cleavage to produce the VP8* and VP5* subunits. These regions contain four (8-1 to 8-4) and five (5-1 to 5-5) antigenic epitopes, respectively [32,33]. The amino acid substitutions between the Rotarix^®^ vaccine strain and Mozambican P[8] strains were concentrated in the 8-1 and 8-3 epitopes. There were five conserved amino acid substitutions, at positions E150D, N195D/G, S125N, S131R and N135D. Sporadic mutations were observed in MAN0033 (N195S and N113D), HJM0338 (S146N) and HGM1789 (P114T) (Figure 4).

## 3. Discussion

In the present study, whole genome sequencing was performed for 36 G1P[8] RVA strains obtained from Mozambican children with gastroenteritis between 2012–2017 (12 from the pre-vaccine period and 24 from the post-vaccine period). This is the first study to perform whole genome analysis of G1P[8] strains in Mozambique, facilitating the description of genetic diversity and the origins of Mozambican strains. 

Of the 36 strains characterized, 33 clustered within the same conserved Mozambican clade across all trees. Two strains, HGM0544 and HJM1646, showed varied patterns by clustering within and were distinct from the Mozambican clade across trees, suggesting these strains had undergone reassortment events. The strain, MAN0033, clustered distinctly from the rest of the Mozambican strains in all trees. This strain was closely related to a conserved group of Malawian G1P[8] strains suggesting that this strain may have been recently introduced from a neighboring country. No distinct clustering patterns were observed based on the year of isolation or vaccination status, which suggests that strains with limited sequence diversity may have circulated among children in the country over the five year period investigated (2012–2017). The homogeneous population of G1P[8] strains suggests that the introduction of Rotarix® has not resulted in a dramatic shift in the diversity of G1P[8] strains circulating in Mozambique. A similar finding was reported in South Africa where no distinct clustering was observed for strains from the pre- and post-vaccine introduction period [26]. Analysis of G1P[8] strains in Brazil over a 27 year period also did not detect any evidence of a selective pressure exerted by the mass introduction of Rotarix^®^ [34]. In contrast, Australia and Belgium reported some unique clusters of G1P[8] strains following vaccine introduction, which may have been due to natural fluctuation or the first signs of vaccine-driven evolution [35]. In Rwanda, unique clusters of G1P[8] strains were identified following RotaTeq introduction [36]. Although neighboring countries reported the widespread (Malawi) and sporadic (South Africa) detection of G1P[8] strains that had undergone reassortment with DS-1 like strains [28,37], all Mozambican strains characterized in this study exhibited a typical Wa-like genetic backbone. Despite some reports of vaccine-derived G1P[8] strains detected in Australia and England, none of the strains identified in this study were derived from the Rotarix^®^ vaccine [38,39]. 

Although there are seven recognized lineages described for global G1 sequences [40], the majority of Mozambican strains from this study clustered in lineage II, with only two strains clustering in lineage I. However, one strain that clustered in lineage I represented the oldest Mozambican strain sequenced in this study from 2012, which clustered with previously characterized G1P[8] Mozambican strains from 2011 [14]. This may suggest that lineage I strains were replaced in later years by G1 strains associated with lineage II [26,41]. The VP7 lineage I strains were detected in the south of Mozambique which may suggest geographical restriction in the circulation of strains. However, these results can be in part due to the short sampling period of this study. Of the four established lineages of the P[8] genotype [21], all Mozambican strains clustered in lineage III and shared a high level of genetic similarity, except strain MAN0033 which clustered in a distinct sub-lineage. 

Maximum likelihood phylogenetic analysis showed that the majority of the Mozambican strains, with the exception of MAN0033, were most closely related to a conserved group of Indian strains across most genes. Even the two reassortant strains (HJM1646 and HGM0544) were most closely related to Indian strains. This suggests that there may have been multiple, contemporary introductions of diverse strains from a similar origin, perhaps India, into Mozambique. This was further supported by the results of the Bayesian analysis, where the time to the most recent common ancestor for the VP7 and VP4 genes of the main Mozambican clade were 2011.7 and 2013.0, respectively, and which had diverged from the closest Indian strain in 2011.7 and 2011.6, respectively. This suggests that the strains became endemic shortly after being introduced and became the dominant variant circulating in the population. These Indian strains were submitted directly to the GenBank database and no associated manuscripts were found, so it is unclear if these strains were associated with any particular outbreak or were detected as part of routine surveillance. 

Overall, the VP7 and VP4 antigenic epitopes exhibited conserved substitutions among the Mozambican strains when compared to Rotarix^®^. The substitutions in the 7-2 VP7 epitope at position M217I, N147D and 8-1, 8-3 VP4 at positions E150D, N195D and S125N, S131R, N135D were observed in pre- and post-vaccine introduction strains, suggesting that these substitutions are not due to the vaccine introduction. 

The main limitation of the study was the limited number of strains successfully sequenced, and that fewer strains were sequenced from the pre-vaccine period.

There is a need to expand the whole genome analysis to other strains detected in Mozambique such as G3, G9 in combination with P[4], P[6] and P[8] genotypes reported previously [18] in order to evaluate the possible influence of vaccine introduction on other rotavirus genotypes. 

Mozambique has introduced the Rotarix^®^ vaccine, however cases of rotavirus infection associated with G1P[8] strains resulting in hospitalization of children are still being reported. The present analysis showed that G1P[8] strains detected in the post-vaccine period did not undergo significant mutations in the epitope regions that could result in vaccine escape. However, the short post vaccine period analyzed (three years) may have influenced these results, as it may be too early to see major genetic changes associated with vaccine pressure. These results highlight the need for future studies to understand host factors such as the role of histo-blood group antigen status, nutritional status and enteric co-infections that can influence the vaccine effectiveness in Mozambique. 

## 4. Materials and Methods 

### 4.1. Ethics Approval 

The ViNaDia Protocol was approved by the National Health Bioethics Committee of Mozambique (CNBS) under number (IRB00002657, reference Nr: 348/CNBS/13). Participants’ anonymity and confidentiality were guaranteed.

### 4.2. Sample Collection 

Forty-three fecal samples, collected between 2012 and 2017 that were positive for RVA by ELISA (Prospect EIA rotavirus, Basingstoke, UK) and identified as genotype G1P[8] by multiplex RT-PCR according to described protocols [42,43], were selected for sequencing according to year of isolation, location of collection (region of Mozambique) and the child’s vaccination status. The samples were obtained from children <5 years of age, hospitalized with acute gastroenteritis, and collected at five sentinel sites of the National Diarrheal Surveillance (ViNaDia), which are Hospital Geral de Mavalane (HGM), Hospital Geral Jose Macamo (HJM), Hospital Central da Beira (HCB), Hospital Geral de Quelimane (HGQ) and Hospital Central de Nampula (HCN), and from a previous study of Centro de Investigação em Saúde da Manhiça (CISM) in Mozambique between 2012 and 2013 (Appendix A) [15]. Clinical information was collected through a structured questionnaire from ViNaDia which included metadata such as age, gender, site and vaccination status.

### 4.3. RNA Extraction and cDNA Synthesis

Total RNA was extracted from stool samples with TRI-reagent (Sigma, Darmstadt, Germany) and single-stranded RNA was precipitated with lithium chloride. The self-priming PC3-T7 loop primer (Integrated DNA Technologies, Coralville, IA, USA) was ligated to dsRNA in order to obtain full-length sequences and cDNA was synthesized using the Maxima H Minus double-stranded cDNA kit (Thermo Fisher Scientific, Massachusetts, MA, USA) as previously described [20,44]. 

### 4.4. Next Generation Sequencing 

The whole genome sequencing was performed using an Illumina MiSeq sequencing platform (Illumina, Inc. San Diego, CA, USA) at the Next Generation Sequencing Unit at the University of the Free State (NGS-UFS). Sequencing was completed using the Nextera XT DNA Library Preparation Kit (Illumina, Inc., San Diego, CA, USA) using protocols previously describe [19].

### 4.5. Data Analyses 

A de novo assembly was performed for all samples using CLC Bio Genomics Workbench (12.0.3; Qiagen, Aarhus, Denmark); all contigs with an average coverage above 100 were identified on the Nucleotide Basic Local Alignment Search Tool (BLASTn at the National Center for Biotechnology information (NCBI). References were chosen based on the Blastn results for reference mapping and extraction of consensus sequences for each segment. The genotyping tools, Virus Pathogenic database and analysis resource (ViPR) [45] and RotaC v2.0 [46], were used to determine the genotype of each gene. The sequences were submitted to GenBank and accession numbers MT737379-MT737772 were assigned.

### 4.6. Phylogenetic Analysis 

Multiple nucleotide sequence alignments with strains obtained from GenBank [24] were made with multiple sequence alignment program (MAFFT v7.450) [47] on Geneious prime v2020.0.3 and Multiple Sequence Comparison by Log Expectation (MUSCLE) [48] alignment available in Molecular Evolutionary Genetic Analysis X (MEGA X) [29]. The optimal nucleotide substitution model for phylogenetic analysis was selected based upon the Akaike information criterion (corrected) (AICc) ranking implemented in the model selection algorithm available on JModelTest [30] and the models selected for each segment were: Tamura-3 parameter (T92+G+I) [49] for VP7; General Time Reversible (GTR+G+I) [50] for VP3; GTR+G for VP2, VP4, NSP2 and NSP3; Hasegawa Kishino Yano (HKY+G+I) for VP6 and NSP1; and HKY+G for VP1, NSP4 and NSP5/6 [51]. The maximum-likelihood trees were generated using MEGA X [29] using 1000 bootstrap replicates to estimate branch support. Pairwise distance matrix nucleotides were obtained in MEGA X using the p-distance algorithm [29]. Amino acid sequences of the VP7 and VP4 Mozambican strains were aligned and epitopes were identified and compared to those of the vaccine strain Rotarix^®^ (A41CB052A, with accession numbers JN849114 and JN849113 for VP7 and VP4) using MEGA X [23,31].

### 4.7. Evolutionary Analysis 

Maximum likelihood trees were generated using the Randomized Accelerated Maximum Likelihood (RAxML) program (v2.0.0) [52], applying the nucleotide substitution model GTR+G. The trees were used as the input for TempEst 1.5.3 to plot root-to-tip genetic distances, and sequences not conforming to a linear evolutionary pattern were discarded [53]. Time-measured evolutionary histories were reconstructed using the Bayesian Evolutionary Analysis Sampling Trees (BEAST) Program package (v 1.7.5) [54]. The nucleotide substitution model Hasegawa-Kishino-Yano model (HKY+G) for VP7 and GTR+G for VP4 were selected based on AICc raking in jModelTest [30].

The parameters applied included a relaxed uncorrelated lognormal molecular clock to account for varied evolutionary rates among lineages and a coalescent Gaussian Markov random field (GMRF) Bayesian Skyride tree prior. Three independent Markov chain Monte Carlo (MCMC) chains were run for 200 million generations with sampling every 20,000 generations, with the first 10% discarded as burn-in. Convergence and mixing of the chains was assessed using Tracer (v1.7.1) and all parameters yielded effective sample sizes ≥ 200 [55]. The Maximum Clade Credibility (MCC) trees were summarized using TreeAnnotator (v1.10.4) [54]. The time-ordered MCC trees were visualized in FigTree (v1.4.4) (http://tree.bio.ed.ac.uk/software/figtree/).

## 5. Conclusions

This study provides important insights into the whole genome sequences of G1P[8] strains in Mozambique. Whilst similar strains were detected prior to and following vaccine introduction, multiple introductions of diverse strains from India highlight the importance of continuously monitoring the strains detected in Mozambique to determine if the strains are evolving by vaccine-induced selection or by natural evolutionary pressures.

## Figures and Tables

**Figure 1 pathogens-09-01026-f001:**
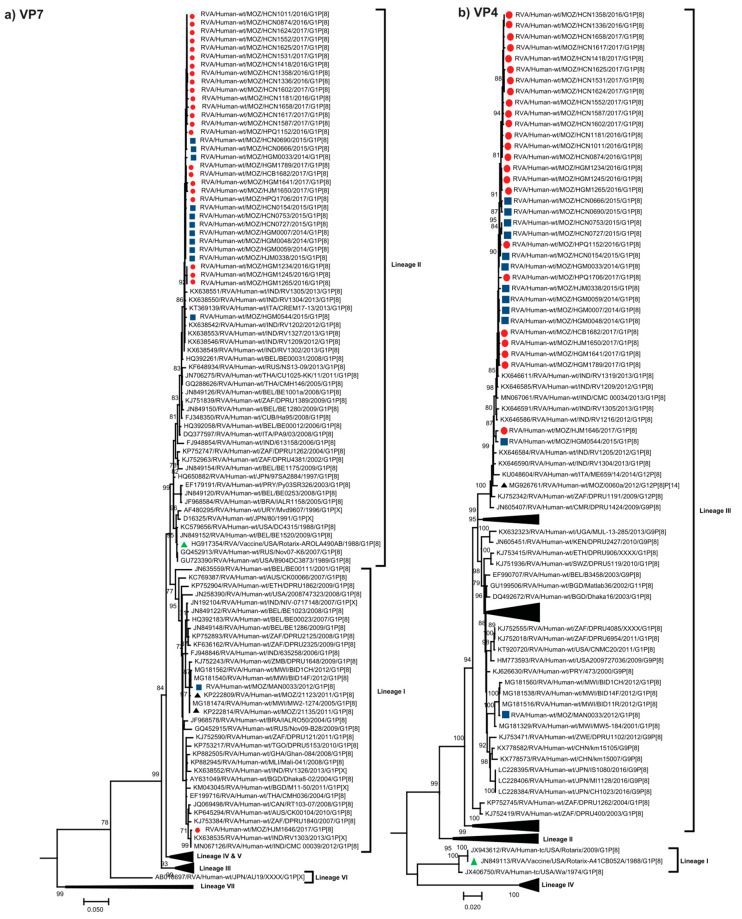
Phylogenetic trees based on the ORF (open reading frame) nucleotide sequence of the (**a**) VP7 and (**b**) VP4 genes of G1P[8] strains circulating in Mozambique and global strains obtained from GenBank. The trees were constructed based on the maximum likelihood method implemented in MEGA X [29], applying the best-fit nucleotide substitution model Tamura-3-parameter (T92+G+I) for VP7 and General Time Reversible (GTR-G) for VP4, determined by JModelTest [30]. Bootstrap values (1000 replicates) ≥70% are shown with DS-1 serving as an out-group (not shown in the final tree). Scale bar indicates genetic distance expressed as the number of nucleotide substitutions per site. Pre-vaccine Mozambican strains are indicated by blue squares, post-vaccine by red circles, the Rotarix^®^ vaccine strain by a green triangle and Mozambican strains from previous studies [14,19] are indicated by black triangles. Lineages are defined from I-VIII for VP7 and I-IV for VP4 [21,22,23,25,26,27,28].

**Figure 2 pathogens-09-01026-f002:**
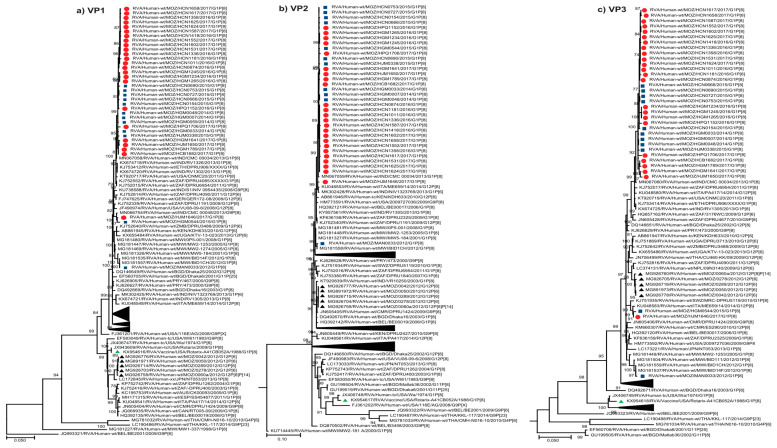
Phylogenetic trees based on the ORF nucleotide sequences of the (**a**) VP1, (**b**) VP2, (**c**) VP3, (**d**) VP6, (**e**) NSP1, (**f**) NSP2, (**g**) NSP3, (**h**) NSP4 and (**i**) NSP5 genes of G1P[8] strains circulating in Mozambique and global strains obtained from GenBank. The trees were constructed based on the maximum likelihood method implemented in MEGA X [29], using the best-fit nucleotide substitution model General Time Reversible (GTR+G+I) for VP3, GTR+G for VP2, NSP2 and NSP3, Hasegawa Kishino Yano (HKY+G+I) for VP6 and NSP1, HKY+G for VP1, NSP4 and NSP5/6, determined by JModelTest [30]. Bootstrap values (1000 replicates) ≥70% are shown with DS-1 serving as an out-group (not shown in the final tree). Scale bar indicates genetic distance expressed as the number of nucleotide substitutions per site. Pre-vaccine Mozambican strains are indicated by blue squares, post-vaccine by red circles, the Rotarix^®^ vaccine strain by a green triangle and Mozambican strains from a previous study [19] are indicated by black triangles.

**Figure 3 pathogens-09-01026-f003:**
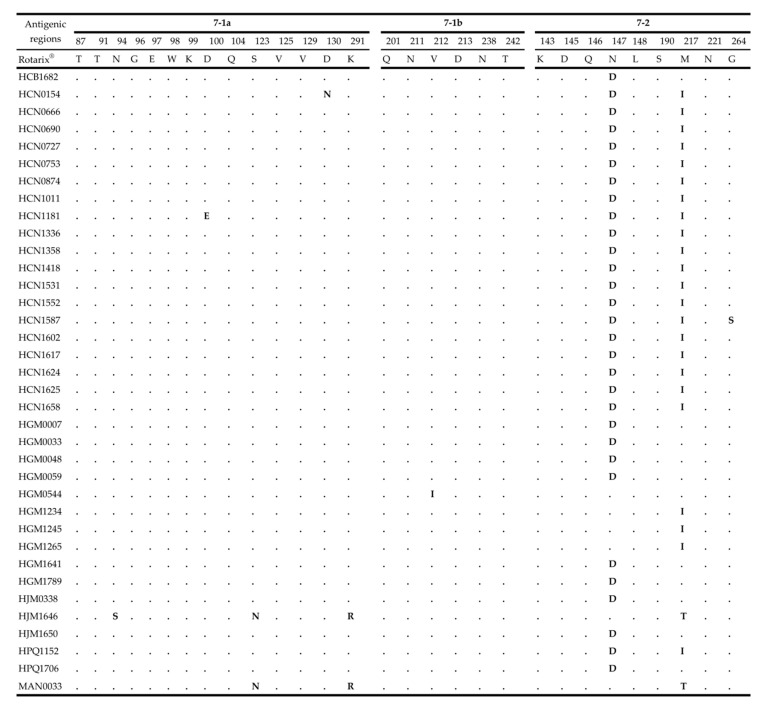
The alignment of amino acids corresponding to three VP7 antigenic epitopes (7-1a, 7-1b and 7-2). The amino acid sequence of Rotarix^®^ is the reference strain and the conserved residues between the Rotarix^®^ to Mozambican strains are indicated by dots (.) and residues that differ are in bold.

**Figure 4 pathogens-09-01026-f004:**
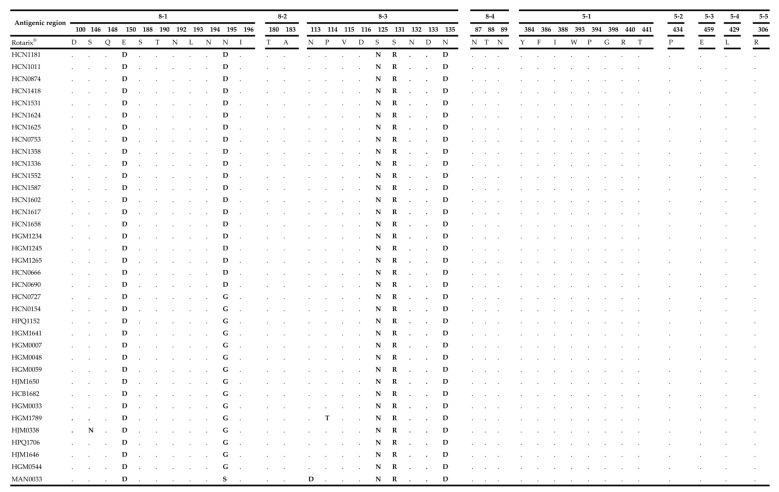
The alignment of the amino acids corresponding to the VP4 antigenic epitopes (8-1, 8-2, 8-3, 8-4 for VP8* and 5-1, 5-2, 5-3, 5-4, 5-5 for VP5). The amino acid sequence of Rotarix^®^ is the reference strain and the conserved residues between the Rotarix^®^ to Mozambican strains are indicated by dots (.) and residues that differ are in bold.

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
