# Peer review of "Whole Genome Characterization and Evolutionary Analysis of G1P[8] Rotavirus A Strains during the Pre- and Post-Vaccine Periods in Mozambique (2012–2017)"

_pathogens, 2020, doi:10.3390/pathogens9121026_

Round 1

Reviewer 1 Report

The manuscript by Munlela and colleagues describes the genetic characterization of whole genome of G1P8 rotavirus strains from Mozambique. The study compares a limited number of strains (n=36) collected pre and post rotavirus vaccine introduction. The phylogenetic description of the strains is very well described and all contained the WA-like genome segments. Some minor sequence changes were reported in many of the genes, and clustered in the same clades. The authors contend that their analysis shows that the vaccine has no selective pressure on the circulating strains. However this comment maybe too early. The authors should provide a description of the rotavirus vaccine uptake, what percentage of the population have received vaccine and how quickly was it taken up in the 3 years post vaccine introduction in 2015. In the context of vaccine pressure on G1P8 strains the timing could be too early. The authors do provide a small glimpse into existence of vaccine pressure where they indicate that changes in the circulating strains has been observed post vaccine introduction.    

Very few strains (n=9) were analyzed prevaccine. How representative of the G1P8 strains were these strains? Was there any other data that could be used to illustrate these were representative of a large cohort of strains - eg elecropherotypes?

when searching for examples of vaccine pressure may researchers seek large changes such as whole genome segment shifts, however subtle changes in gene sequence maybe more illustrative of vaccine pressure. Infact the data does suggest that many of the non structural gene segments may have a small number of differences with several of the post vaccine strains (eg NSP2, NSP3 NSP4), these could be the first illustration of selective pressure. I commend the authors for looking at the antigenic regions for alteration which could reflect immunological alterations. The change at position 147 of VP7 - N147D - did this result in a loss of a potential carbohydrate attachment site in theses strains?

The major challenge with the study is that the timing maybe too early to see much, that is 3 years post introduction might be too early to clearly see genetic changes. In addition the baseline data set is extremely limited, so given these constraints it may be difficult to really identify whether vaccine pressure has been exerted in this setting. However the data provided is extremely good and does provide a very good start.  

I have no specific comments as manuscript is very well written and clear.  

Reviewer 2 Report

The Authors performed the whole genomic analysis of 36 Mozambican G1P[8] rotaviruses collected in pre- and post-vaccine era. All strains showed a Wa-like genetic backbone and the vast majority of the strains tested (33/36)  showed to be closely related to each other and to Indian strains. Evolutionary analyses of VP7 and VP4 suggested a possible introduction of strains from India in a recent time. However, a limitation of this study is the limited number of sequences analyzed of pre-vaccine era (9), this should be focused in the discussion. Phylogenetic and amino acid sequence analyses showed the absence of vaccine selective pressure on rotavirus strains in Mozambique.

The presented data are useful to monitor the genetic evolution of rotavirus strains worldwide in order to value the vaccine performance. However, the manuscript in some part (introduction and results) is too repetitive and therefore could be shortened allowing the text to be more readable.

Scientific English has to be used.

Major revisions:

The title is too long has to be shortened.

Introduction:

Lines 53-77 could be shortened.

Line 77 lineage classification could be added, as lineages are included in phylogenetic analyses.

Results:

Most of data described are repeated in several sections (2.1.1-2.2.3). The majority of  G1P[8] strains (33) clustered in conserved clade closely related to Indian strains, while HJM1646 and HGM0544 are more divergent in several segments and the MAN0033 just in few genes. Revise the result section allowing the reader an   easer acquisition of the data.

The nucleotide and amino acid identity among G1P[8] strains of the study and respect to vaccine strains could be translate in a table and removed from the text (page 3 line 108-113; page 5 lines 173-175 and 194-196).

References has to be checked (i.e.  the reference 21 is repeated as 30)

Minor revisions:

Page 9 lines 221-“these strains…to 2003.5)”, avoid comment into results section.

Page 9 line 244: “however, most of amino acid substitutions were observed”

The reference 23 could be removed

Modifications to the manuscript should be applied accordingly.

Reviewer 3 Report

The article presented by Munlela and coworkers describes the whole genome sequencing of several G1P[8] rotavirus strains from Mozambique. Some of the samples were collected previously to the introduction of the rotavirus vaccine in the country.

The article is well written and easy to follow. The introduction section is enough to understand the problem proposed and the approach followed. The hypothesis of the work is that the introduction of the vaccine has an impact in the G1P[8] strains found in vaccinated children. The authors applied NGS and successfully fully sequenced 36 strains. The phylogenetic and evolutionary analyses have been performed correctly and the results are clearly presented.

The conclusion is that the vaccine has reduced the incidence of rotavirus in the country, but it has not created a shift in the evolutionary history of the virus in the area. Similar results and conclusions have been observed in other regions of the world.

Round 2

Reviewer 2 Report

The title could be furthe modified as: "Whole genome characterization and evolutionary analysis of G1P[8] rotavirus strains in pre- and  post-vaccine period in Mozambique (2012-2017).

Author Response

Revised submission of manuscript pathogens-996311 to Pathogens Journal, Special Issue "Rotaviruses and Rotavirus Vaccines"

The authors thanks the reviewer for the constructive comment and wish to submit our revised manuscript entitled ‘’Whole genome characterization and evolutionary analysis of G1P[8] rotavirus strains circulating in children under five years of age pre- and post-vaccine introduction in Mozambique (2012-2017)’’ by Benilde Munlela, Eva D. João, Celeste M. Donato, Amy Strydom, Simone S. Boene, Assucênio Chissaque, Adilson F. L. Bauhofer, Jerónimo Langa, Marta Cassocera, Idalécia Cossa-Moiane, Jorfélia J. Chilaúle, Hester G. O’Neill8 & Nilsa de Deus.

The authors’ response are in red text and changes to the text are indicated in green.

Kindly find bellow the question response:

Reviewer 2

The title could be further modified as: "Whole genome characterization and evolutionary analysis of G1P[8] rotavirus strains in pre- and  post-vaccine period in Mozambique (2012-2017).

Response: The authors thanks the reviewer for the comment. The title was updated as suggested with slight modification, Page 1 Lines 2-6:"Whole genome characterization and evolutionary analysis of G1P[8] Rotavirus A strains during the pre- and  post-vaccine periods in Mozambique (2012-2017)’’.